# Advances in the Therapeutic Applications of Plant-Derived Exosomes in the Treatment of Inflammatory Diseases

**DOI:** 10.3390/biomedicines11061554

**Published:** 2023-05-27

**Authors:** Xiaofang Wei, Xiuyu Li, Yuejun Zhang, Jian Wang, Shuibao Shen

**Affiliations:** 1College of Animal Science and Technology, Guangxi University, Nanning 530004, China; 2218402005@st.gxu.edu.cn (X.W.);; 2Guangyuan Academy of Agricultural Sciences, Guangyuan 628017, China; 3State Key Laboratory for Conservation and Utilization of Subtropical Agro-Bioresources, Guangxi Key Laboratory of Animal Breeding and Disease Control, College of Animal Science and Technology, Guangxi University, Nanning 530004, China

**Keywords:** plant-derived exosomes, inflammatory disease, plasma-membrane, biogenesis, immune response

## Abstract

Plant-derived exosomes (PLDEs) are small extracellular vesicles that encapsulate proteins, nucleic acids and lipids, and they are usually involved in intercellular communication and molecular transport in plants. PLDEs are widely used in the therapy of diseases due to their abundance and easy availability. The diverse roles of PLDEs, which include transportation of drugs, acting as biomarkers for diagnosis of diseases and their roles in different therapies, suggest that there is a need to fully understand all the mechanisms involved in order to provide the optimum conditions for their therapeutic use. This review summarizes the biogenesis, components and functions of PLDEs and focuses on their use as therapeutic agents in the treatment of inflammatory diseases. It also explores new ideas for novel approaches in which PLDEs could potentially help patients with inflammatory diseases in the future.

## 1. Introduction

Exosomes are 40–160 nm extracellular vesicles with a phospholipid bilayer structure that are secreted by cells [1]. Exosomes mainly function in intercellular communication, molecular transport and in the immune response, and they contain proteins, RNAs and DNAs and lipids [2]. Plant-derived exosomes (PLDEs) were first reported in a study of fungal invasion of barley leaves [3]. The structure of PLDEs is almost similar to that of mammalian-derived exosomes (MDEs), but there are specific differences with regards to the cell source and their functions [4,5]. Different types of extracellular vesicles with specific proteins, lipid components and transcription enzymes have been extracted from various plants [6,7]. PLDEs can mediate cellular signaling [8], stimulate an immune response [9], transport substances between cells [10], and they connect unrelated species such as plants to parasites [11] as well as insects to plants [12]. Due to their diverse roles in regulating biological processes, PLDEs have found an additional role in clinical medicine, where they can act as therapeutic agents in the treatment of a variety of diseases including inflammation [13]. For example, PLDEs can combine with anti-inflammatory cytokines in vivo, thereby acting as an inflammatory immune regulator in order to treat inflammation [14].

As natural carriers, PLDEs can be used to load and transport drugs to the site of inflammation for targeted therapy, as has been the case for treatment of human colitis [15]. In a rat inflammation model, the negatively charged PLDEs were attracted to the positively charged inflammatory area, where they induced the expression of macrophage IL-10 and heme oxygenase-1 (HO-1) at the diseased site. This significantly suppressed production of the pro-inflammatory cytokine, tumor necrosis factor (TNF)-α [16]. PLDEs from plants such as blueberry and shiitake mushroom can mediate the process of inflammation in the body by their differential expression of miRNAs, which have specific anti-inflammatory effects [17]. Therefore, research on PLDEs as therapeutic agents for inflammatory diseases has mainly focused on their use as drug delivery carriers and biomarkers, as well as their ability to induce responses against inflammation [18]. The wide variety of PLDEs, based on their source plant species, component functions, biological activities and specific participation in inflammatory mechanisms, still need further exploration.

In this review, we discuss the biogenesis, composition and functional analysis of PLDEs, as well as their prospects in treating inflammatory diseases. We focused on the efficacy and the associated molecular mechanisms of PLDEs in treating inflammatory diseases. In the referenced publication, the term ‘exosome’ was used when it was clearly referred to, while the term ‘PLDE’ was used when the differentiation was unclear.

## 2. Overview of PLDEs

### 2.1. The Biogenesis of PLDEs

Almost all cells can secrete exosomes, and these are abundant in all living organisms, leading to a very broad range of applications and research directions for exosomes. In order to systematically describe the synthesis and applications of PLDEs for treatment of inflammatory diseases, it is necessary to understand their biogenesis, composition and functions.

The biogenesis of PLDEs is a highly regulated process. PLDEs enter the cell interior after vesicle budding, and they are released as vesicular endosomes from the plasma membrane. The exosomes bud from the plasma membrane in three ways. Firstly, the individual endosomes are transformed during vesicle development into mature multi-vesicular bodies (MVBs), and the resulting exosomes are released when the plasma membrane fuses. Secondly, the exosomes are directly released by vesicles budding from the plasma membrane. Thirdly, they bud directly from the intracellular plasma membrane-connected compartments (IPMC) after these have become dysregulated [19]. The budding efficiency of these three methods is different.

By using atomic force microscopy of stem cells, it was confirmed that the budding rate of exosomes on the plasma membrane of cells occurs at the same rate as the cells are able to produce exosomes [20]. Being rich in lipids, cytosolic and specific proteins, the resultant MVBs have multiple compartments, consisting of several intraluminal vesicles (ILVs). These ILVs mature into exosomes that are primarily secreted outside the cell via the endosomal sorting complex required for transport (ESCRT) [21]. The secretion-driven mechanisms of exosomes are the recognized pathways involving ESCRT [22] and tetraspanin [23]. ESCRT proteins, including Hrs, STAM1 and TSG101, are crucial for exosome biosynthesis and rely on ubiquitin-binding for their function [24]. There are a large number of ubiquitinated soluble compounds, such as MHC-II, in cells, and their presence enhances the enrichment of ILVs [25]. However, non-ubiquitinated MHC-II has also been detected in secreted exosomes [26].

Studies have confirmed that Rab31 and GTPase function as regulators of lipid-mediated exosome biogenesis which are independent of ESCRT and tetraspanin pathways [27]. However, we mainly discuss exosome secretion driven by the ESCRT mechanism, which is composed of a cytoplasmic multi-subunit system that is necessary for membrane remodeling. This enables the vesicles to bud and their constituents to be sorted into MVBs, which relies on the four core ESCRT complexes (ESCRT-0, ESCRT-I, ESCRT-II and ESCRT-III) and ESCRT-related proteins, including TSG101, VPS4 and ALIX [28]. ESCRT plays a key role in exosome biogenesis through the formation of ILVs. Their PLD2-phosphatidic acid aggregates allow the endosomes to enter the interior of cells through ESCRT-I, which then promotes secretion of exosomes from corresponding cells [29].

The interaction between the MVB12B-MABP structure and PLD2-phosphatidic acid promotes exosome secretion as well as disruption of the integrity of the MM (MVB12B-MABP)-PPA pathway. This affects late budding of endosomes which significantly reduces the number of exosomes secreted [29]. Compared with the traditional ESCRT pathway, when the transmembrane proteins of intracellular MVBs and late endosomes are transported in vitro, ESCRT-III is aggregated by ALIX into late endosomes [30]. This promotes the extracellular secretion of transmembrane proteins through the cell membrane. In this process, ESCRT-III directly interacts with blood bisphosphatidic acid (LBPA) and bypasses the complex ESCRT process [31].

### 2.2. The Molecular Composition of PLDEs

The structures of PLDEs are similar to those of animal exosomes. The main components of PLDEs are lipids, proteins and nucleic acids. The specificity of PLDEs mainly depends on their cellular origin. These three substances are the key factors [32,33], and each component of exosomes has its corresponding function. In general, the protein profiles will determine the uptake mechanism: lipids are required for efficient cellular uptake and the nucleic acids will function in recipient cells [34].

The absorption of exosomes is mainly related to their lipid composition. PLDEs are enriched in phospholipids, such as phosphatidic acids (PA), phosphatidylethanolamines (PE) and phosphatidylcholine (PC) [35]. The commonest lipid in PDLEs is PA, and it is a key component for exosome delivery; for example, PA was found to regulate their pathogenicity by affecting the interactions between exosomes and hemopexin in periodontal pathogenic bacteria [36]. Other studies have shown that PC-enriched PLDEs can be preferentially absorbed by Ruminococcus, showing that the specific lipid composition is an important factor affecting the targeted uptake of exosomes by cells [37]. The importance of absorption of lipid composition was shown when it was found that modified nanoparticles could be effectively absorbed by mouse intestinal cells [38]. Imaging analysis showed that this was due to the smaller size and PA content of the edible-labeled PLDEs in murine enterocytes, which resulted in the nanoparticles being formed in the intestines [39]. Therefore, the lipid composition of PLDEs can be modified and assembled into nanocarriers with novel structures [40], which can aid in their application as drug carriers.

The protein content will determine the function of PLDEs. PLDEs can serve as signal transductors through protein kinases and G protein, intracellular transporters through annexins, exosome biogenesis through ubiquitin, clathrin and ALIX, as well as cell adhesion through an integrin such as lactadherin [41]. Animal and PLDEs are rich in proteins, and their functions and operating mechanisms are different. In humans, CD63 is considered to be the most common protein biomarker for exosomes, and it has been used as an important marker in cancer cell research [42]. In plants, SYP121 is a plant-specific syntaxin protein that is resistant to fungal pathogens [43]. When plant leaves are infected by pathogens, PLDE secretion in the leaves will increase, allowing their cellular uptake. The amount of digested GDVSs is significantly lower than that of undigested GDVs, and the digestion of GDVs is highly correlated with the combination of type II lectin, so preventing the combination of GDVs and type II lectin will reduce the internalization of GDVs in cells. During the process, the protein sites on the surface of GDVss can bind to cd98 and type II lectin, so GDVss can reduce the binding to type II lectin by binding to cd98, thereby promoting the ability of GDVs itself to be taken [44].

PLDEs usually contain a large number of non-coding RNAs, which are involved in various cell activities, such as cell proliferation, apoptosis, metabolism and immune responses by controlling gene expression in receptor cells [45]. For instance, NanSight tracking system and electron microscopy have revealed that miR-156a, miR-168a and miR-166a in exogenous PLDEs were absorbed by rat intestinal epithelial cells (IEC6) and the expression of TNF-α in mammalian fat cells were affected by them [46]. However, it cannot be confirmed in humans. This demonstrated that PLDE could be used as drug delivery carriers for gene therapy [47]. The argonaute (AGO) protein of eukaryotes can selectively bind to siRNAs and guide the complex to a specific gene site by targeting the base complementation between a mRNA and the siRNA. Subsequent studies showed that AGO protein can selectively bind to PLDE-rich cellular sRNA in Arabidopsis [48]. There are 10 subtypes of AGO proteins, and all have different siRNA binding properties. Arabidopsis PLDEs, with its related siRNAs, were only detected to bind to AGO1 protein [49], showing that the siRNAs affected the protein metabolism of the PLDEs. The siRNAs in PLDEs were also shown to promote the secretion of IL-23 and TNF-α from macrophages infected with Helicobacter pylori in order to induce an inflammatory response in target cells [50]. This occurred by reducing the expression of specific antigenic genes and maintaining cell homeostasis [51].

Increasing evidence has revealed that PLDEs can be absorbed by animals and the components such as proteins, lipids and nuclear acids could function in animal cells; however, there are still some differences between PLDEs and mammal-derived exosomes (MDEs). For example, an obvious difference between PLDEs and MDEs is that PLDEs are enriched in phospholipids, such as phosphatidic acids (PA), phosphatidylethanolamines (PE) and phosphatidylcholine (PC), while MDEs are composed of cholesterol and sphingomyelins [35]. Here, we summarized the advantages and disadvantages of PLDEs and MDEs in Table 1 and Table 2.

## 3. Therapeutic Targeting of PLDEs in Inflammatory Diseases

### 3.1. Lung Inflammation

Exosomes, which are produced by endogenous alveolar epithelial cells, macrophages, pulmonary microvascular endothelial cells and neutrophils have been found to regulate immunity in cases where lungs are damaged and inflamed [67]. In recent years, PLDEs have been thought to communicate with animal cells across many species, and their role in immune regulation cannot be ignored. A compromised immune system and inflammation in the lungs are hallmarks of coronavirus disease 2019 (COVID-19) infection [68]. In a study involving mice exposed to severe acute respiratory syndrome coronavirus 2 (SARS-CoV-2), researchers found that exosomal Nsp12 and Nsp13 activated nuclear factor kappa B (NF-κB) in lung macrophages, leading to the expression of inflammatory cytokines such as TNF-α, interleukin (IL)-6 and IL-1β (as shown in Figure 1A). Ginger exosome-like nanoparticles (GELN) were selected for tracheal delivery therapy, and miRNA aly-miR396a-5p was efficiently delivered to the lungs by this method. The expression of the viral S and Nsp12 was inhibited by GELN miRNAs, which resulted in the inhibition of cytopathic effects (CPEs) observed in SARS-CoV-2-infected Vero E6 cells. It was also found that Nsp12 exposure alone did not activate NF-κB, but a targeted delivery of ginger miRNAs to lung epithelial cells and macrophages could inhibit the expression of Nsp12, thereby preventing exosomal Nsp12-mediated lung inflammation. GELN could also inhibit the production of TNF-α, IL-6 and IL-1β and achieve its purpose of prevention and treatment by inhibiting the NF-κB pathway [69]. It is worth noting that viral exosomes during disease progression induce a series of inflammatory cytokines that promote inflammation, and GELN can counteract this by inducing endogenous exosomes. This example shows that PLDEs can have a relationship with endogenous host exosomes, but the specific mechanisms of these reactions still need further clarification.

Traditional Chinese medicines used in the treatment of lung diseases are usually taken in a decoction produced by boiling herbs in water, and the active ingredients in the extracts are usually stable. ELNs derived from traditional Chinese medicinal plants may also play a role in these decoctions. Both the ELNs extracted from decoctions and those obtained in vitro have showed therapeutic effects in vivo, although those from the latter source had stronger therapeutic effects. ELNs containing siRNAs, specifically HJT-sRNA-m7 and PGY-sRNA-6, have demonstrated strong abilities to combat fibrosis and inflammation, respectively. Based on the two components that play a major role, sphingosine-HJT-sRNA-m7 and sphingosine-PGY-sRNA-6 benzyl bodies co-assembled with sphingine and siRNAs could improve post-oral stroke in mice, respectively. These effects were seen in both bleomycin-induced pulmonary fibrosis and poly(I:C)-induced lung inflammation [70].

### 3.2. Liver Inflammation

Non-alcoholic fatty liver disease (NAFLD) [71], alcoholic liver disease (ALD) [72] and chronic hepatitis B (HBV) [73] are all characterized by hepatic inflammation. The anti-inflammatory mechanisms of hepatic inflammation are a key pathway for drug treatment of liver diseases.

GELP can target liver cells, and its actions may be due to the bioactive compound, gingerol [74], that is carried by ginger-derived nanoparticles (GDN). This is not the free form source of gingerol, and it is delivered directly to liver cells where it is dehydrated to form 6-shogaol, which then activates nuclear factor erythroid 2-related factor 2 (Nrf2) [75]. Nrf2 has anti-inflammatory effects, and in in vitro cell cultures of mouse hepatocytes assays, it was shown that 6-shogaol plays an important role in GDN-mediated Nrf2 activation. This was conducted via the TLR4/TRIF pathway, and it inhibited hepatocyte inflammation, which was shown by using lipid knock-out and knock-in strategies [76].

The roles of surface proteins during endocytosis were confirmed by comparing the uptake of garlic-derived nanovesicles (GDVs) that had lost all these molecules to those that were non-trypsinized. In addition, blockade of the CD98 receptor in the HepG2 cell line significantly reduced GDV uptake. The mannose-specific binding lectin (lectin II) on the surface of GDVs was saturated with mannose, and CD98 antibody blockade of the CD98 receptor was shown to downregulate GDV internalization. In vitro studies have shown that GDVs have anti-inflammatory effects. This is demonstrated by the down-regulation of pro-inflammatory factors such as IFN-γ and IL-6 mRNA levels in HepG2 cells. This may be due to GDVs targeting CD98 in vitro and in vivo and reducing the levels of hepatitis-related pro-inflammatory cytokines, such as TNF-α, IFN-γ and IL-6, thus inhibiting liver inflammation [44].

The liver is prone to attack by pathogenic microorganisms as well as drug toxins. Overactivation of the pyrin domain-containing 3 (NLRP3) inflammasome has been shown to cause inflammation and contribute to the development of diseases, ultimately leading to liver damage. The NLRP3 inflammasome is a high molecular weight multi-protein complex that recruits apoptotic speck protein containing a caspase recruitment domain (ASC) and caspase 1. These molecules play a key role in primary and adaptive immune responses [77]. Previous studies have shown that treatment with ginger rhizome exosome-like nanoparticles (G-ELNs) can inhibit the pathways downstream of inflammasome activation in primary macrophages (Figure 1B). The inhibition of NLRP3 inflammasome assembly and activation is achieved through caspase 1 self-cleavage and subsequent secretion of IL-1β and IL-18 [78].

The study about exosomes of shiitake mushrooms shows that ELNs from various common mushroom species but found that only shiitake-derived ELNs (S-ELNs) significantly inhibited NLRP3 by preventing inflammasome formation in primary macrophages. Inflammasome activation, which occurs via a different mechanism to lentinan, inhibits inflammation by targeting and inhibiting AIM2 inflammasome-associated activity. The results of the immunofluorescence staining indicated a significant reduction in the formation of ASC in cells treated with S-ELN. This suggests that S-ELN targets the initial stages of the NLRP3 inflammasome pathway to prevent its assembly. The study found that pretreatment with S-ELN led to a decrease in pro-IL-1β and NLRP3 in the livers of mice that were exposed to GalN/LPS-induced stress. The results indicate that S-ELN has an inhibitory effect on the secretion of IL-6 and the protein and mRNA expression levels of IL-1β. The results suggest that S-ELN may have the ability to reduce the severity of acute liver injury in mice caused by GalN/LPS [79].

### 3.3. Intestinal Inflammation

The gut is exposed to many bacterial and dietary sources of pathogenic microorganisms, and persistent immune responses against invading pathogens may lead to acute and chronic intestinal inflammatory diseases (IBDs). Inflammatory bowel diseases (IBDs) are classified into two main types—Crohn’s disease and ulcerative colitis. However, the exact causes of these diseases are still unknown and vary widely. Although immunomodulators, anti-TNF drugs and monoclonal antibodies have been developed to treat intestinal diseases, they can result in long treatment cycles, repeated medications, side-effects and high costs for patients. In contrast, natural exosomes isolated from plants such as bitter melon [6], garlic [80] and cannabis [81] which are traditional medicines, can overcome many of these limitations by providing sustainable, safe and economical therapeutic modalities [82].

Purified PLDEs of plant extracts from pineapple, grapefruit and grape that contain miRNAs have been shown to be absorbed by intestinal cells in the mammalian small intestine [47]. This is a promising means of targeting PLDEs to treat intestinal inflammation. Ginger is a medicinal and edible plant whose exosomal vesicles have been shown to have multiple therapeutic benefits in inflammatory diseases. Previous studies found that orally administered GELNs preferentially promoted the secretion of anti-inflammatory cytokines, IL-10 and HO-1 by inducing Nrf2 activation in macrophages and intestinal Wnt/TCF4 activation in mice [83]. Zhang et al. characterized a population of specific GELNs and demonstrated their potent targeting of the colon after oral administration.

Intestinal epithelial cells (IECs) are the first site of invasion after inflammation, and they play a role in intestinal physical defense (Figure 1C). GELNs were mainly absorbed by the IECs and macrophages, and when administered orally, these vesicles increased the survival and proliferation of IECs in a colitis model. Research has demonstrated that they have the ability to reduce the expression of pro-inflammatory cytokines such as TNF-α, IL-6 and IL-1β, while simultaneously increasing the expression of anti-inflammatory cytokines such as IL-10 and IL-22. Therefore, GELNs have the potential to attenuate injury factors while promoting a healing effect during intestinal inflammation [61]. Another study showed that GELNs were also effective in specifically targeting colonic tissue and attenuating colitis by inhibiting the expression of CD98 [84].

In a dextran sodium sulfate-induced colitis model in mice, grapefruit-derived nanovesicles (GRELPs) were selectively taken up by intestinal macrophages through upregulation of HO-1. The mechanism of expression and inhibition of IL-1β and TNF-α production in intestinal macrophages improved colitis symptoms [85]. Grape exosome-like nanoparticles (GrapeELNs) can penetrate the intestinal mucus barrier to act on mouse intestinal stem cells (mISCs), be selectively absorbed and promote the growth of leucine-rich-repeat-containing G-protein-coupled receptor 5(Lgr5) [86] through the Wnt/β-catenin signaling pathway in the mouse intestine proliferation. Based on the GrapeELN-induced expression of mISCs growth genes, these vesicles were shown to promote a dramatic proliferation of mISCs and resulted in mucosal epithelial regeneration and rapid restoration of intestinal architecture in steady-state and injury-induced colitis models in mice [38].

In a mouse model of LPS-induced colitis, orally administered turmeric-derived nanoparticles (TNDPs) were preferentially localized to the inflamed colon and were mainly internalized by colonic epithelial cells and macrophages. In mice, TDNPs were found to alleviate colitis symptoms by regulating the expression of pro-inflammatory cytokines (TNF-α, IL-6 and IL-1β) and the antioxidant gene HO-1, while also deactivating the NF-κB pathway [87]. Zu et al. isolated natural exosomes from tea leaves and made exosome-like nanotherapeutics (ENTs) whose surface galactose groups can mediate the growth of macrophages through galactose receptor-mediated endocytosis by specific internalization. Enteric neurons (ENTs) have been shown to have anti-inflammatory effects on the intestines and colon. This process involves reducing the production of reactive oxygen species, inhibiting the expression of pro-inflammatory cytokines and enhancing the secretion of the anti-inflammatory factor IL-10 by macrophages [82].

In the intestinal tract, PLDEs can not only act on macrophages, intestinal epithelial cells and other cell types to exert immune effects, but they may also exert these through the intestinal flora. For example, ginger is a perennial herb of the ginger genus, which has anti-inflammatory and anti-vomiting effects. Lactobacillus bacteria have a preference for taking up GELNs through surface lipids. These GELNs also contain microRNAs that target different genes found in LGG (Lactobacillus rhamnosus). The targeting of LGG monooxygenase ycnE by GELN mdo-miR7267-3p results in the production of increased levels of indole-3-carbaldehyde (I3A). Both GELN-RNA and I3A act as ligands for aryl hydrocarbon receptors, thereby inducing anti-inflammatory factor IL-22 production, which can relieve colonic inflammation [39]. Exosome-like nanovesicles (MBELNs) extracted from mulberry bark can act on virulent Lis-EGD (Listeria monocytogenes strain, EGD) unique protein, which subsequently inhibits the expression of bacterial mRNA. The MBELNs have the ability cause to Lis-EGD (Listeria (L.) monocytogenes-EGD) species-specific growth inhibition [88] and maintenance of gut microbial homeostasis.

Collectively, PLDEs communicate with host immune cells, intestinal epithelial cells, macrophages and gut microbiota, and their targets and functions are T cells (Tregs). Resistance to them is influenced by various mechanisms such as macrophage maturation and microbial homeostasis. However, they are able to promote the secretion and expression of anti-inflammatory factors and inhibit intestinal inflammation. Here is a meta-analysis about PLDEs functions of different species and applications of PLDEs in anti-inflammatory (Table 3, Figure 2).

## 4. The Inadequacies of PLDEs

The safety of PLDEs continues to be questioned. These doubts mainly focus on the quality stability of plant raw materials and the complex composition of extracts. The safety of PLDEs continues to be questioned, and there are few studies on negative effects. These doubts mainly focus on the quality stability of plant raw materials and the complex composition of extracts [90]. Extracts of PLDEs containing impurities may lead to the spread of the virus in the body [91]. Exosomes from different cells contain different contents and affect other cells [92]. Exosomes are a natural transporter, and exosomes can easily pass through biological barriers, which make exosomes an excellent transport carrier [93]. The biggest advantage of PLDE as a therapeutic agent is that it is rich in types, diverse in functions and has excellent engineering transformation potential as carrier. Many artificially modified exosomes are used as transport carriers to carry siRNA and miRNA into the organism, which requires exosomes carrying genetic material to be very stable and non-invasive to the body except for cellular targets. The stability of many exosomes loaded with siRNA will be reduced, because the modification of targeted substances will destroy the integrity of the exosome biofilm, resulting in extremely fast degradation of exosomes in the systemic circulation, which is of great significance to the purification technology of PLDEs and the loading of target substances. The choice of has extremely high requirements [94,95].

The stability of PLDEs includes the stability of its own structure and the quality of plant sources. The instability of PLDEs is the key reason for the limitation and questioning of the repeatability and expansion of many studies on PLDEs [55].The yield of exosomes is low and many plants can analyze exosomes, but as the number of passages increases, the differentiation ability of stem cells gradually decreases, and the number of apoptotic cells increases significantly from the sixth to the eighth passage [96], which may cause the quality and quantity of PLDEs to be affected [97]. To summarize, there are many similar results that have confirmed that PLDEs can have positive and negative effects on human, but few in-depth ones have been conducted.

## 5. Conclusions and Future Perspectives

The use of PLDEs for the treatment of inflammatory diseases has attracted a lot of interest. Therefore, in recent years, many studies have reported the therapeutic effects and mechanism of exosomes from different plant sources on different inflammation-associated conditions. In this review, we focuse on the direct impact of PLDEs in the treatment of inflammation, including methods for drug delivery, specific cell targets, the secretion of inflammatory factors, the expression of inflammatory genes as well as their immunity-associated effects.

There are some potential techniques that can be used to explore how PLDEs can affect inflammation in animals, such as metabolomics studies, network bioinformatics, clinical trials, synthesis of novel PLDEs using bioactive plant compounds, etc. Metabolomics and network bio-informatics studies have shown that some mRNAs can decrease the inflammatory factors by regulating the expression of related genes in mice. There are many clinical trials using PLDEs in cancer treatments but their use as anti-inflammatory factors is still in their infancy. The use of PLDEs has great potential in medicine.

## Figures and Tables

**Figure 1 biomedicines-11-01554-f001:**
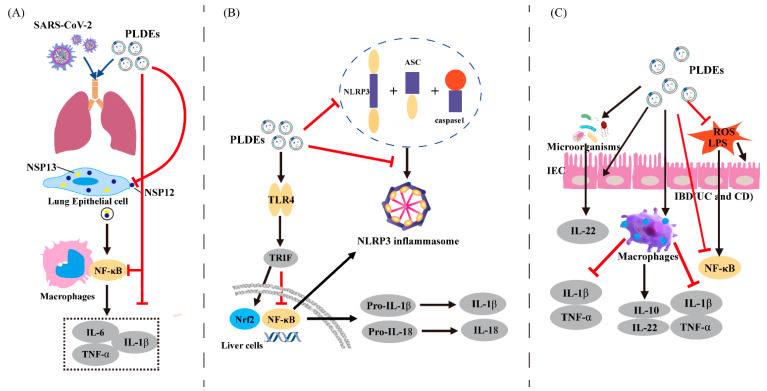
Mechanisms inflammatory diseases. Mechanisms of action of plant-derived exosomes in the treatment of (**A**) pneumonia, (**B**) hepatitis, and (**C**) intestinal inflammation involving PLDEs. The red arrows indicate inhibition, and the black arrows indicate facilitation of specific cellular processes. IBD: inflammatory bowel disease. UC: ulcerative colitis. CD: Crohn’s disease.

**Figure 2 biomedicines-11-01554-f002:**
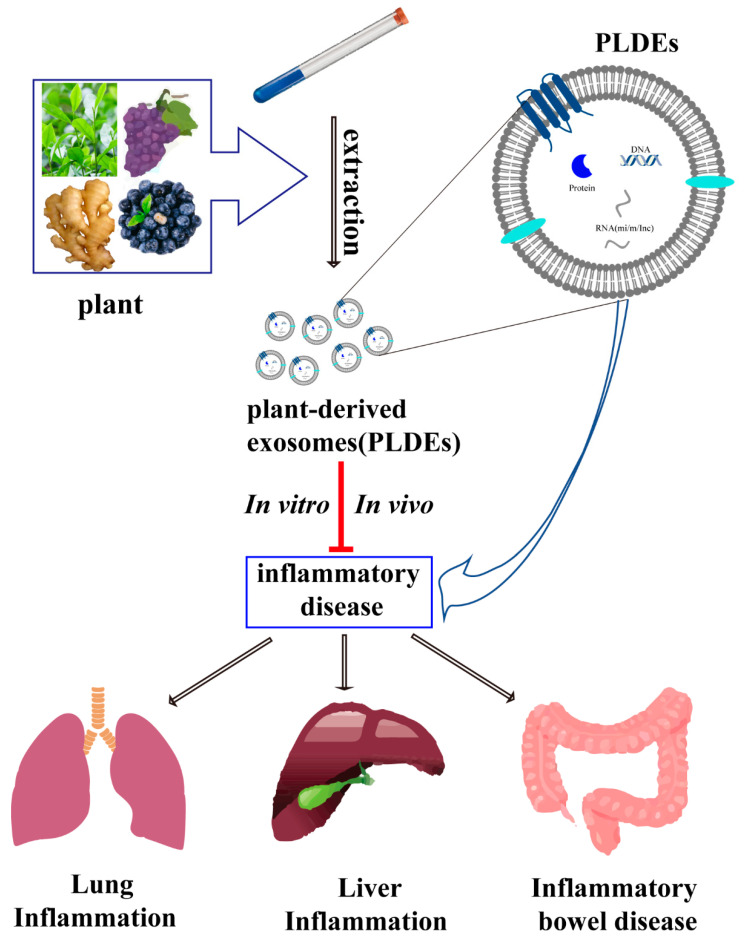
Applications of plant-derived exosomes in the treatment of inflammatory diseases. Plant-derived exosomes (PLDEs) have been extracted from plants such as ginger, mulberry, grape and so on. Previous studies have shown that PLDEs can inhibit inflammatory diseases in liver, lung and intestine from both in vitro and in vivo, which may be due to the action of PLDEs on active molecules such as protein and RNA carried by them.

**Table 1 biomedicines-11-01554-t001:** The advantages and disadvantages of PLDEs on animals.

The Advantages of PLDEs	The Disadvantages of PLDEs
Diverse types and rich sources [52,53]	The database is incomplete, and the identification of specific functions and components is difficult [17,54]
Ease of mass production [35]	There are few clinical trials, and the reliability of research results is low [55]
Relatively high [56]	High dose dependence [34]
High extraction efficiency [57]	Poor stability in digestive system [58]
Excellent drug carriers [59]	
Broad adaptability to inflammatory diseases [60]	
Low toxicity [57]	
High in phospholipids, easily absorbed [38,61]	
Excellent oxidation resistance [62]	
Improve gut microorganism [39]	

**Table 2 biomedicines-11-01554-t002:** The advantages and disadvantages of MDEs on animals.

The Advantages of MDEs	The Disadvantages of MDEs
High stability [63]	The database is incomplete, and the identification of specific functions and components is difficult [17,54]
Less adverse reactions [64]	There are few clinical trials, and the reliability of research results is low [55]
High stability in digestive system [65]	High dose dependence [34]
Great potential for artificial transformation [66]	Poor stability in digestive system [58]

**Table 3 biomedicines-11-01554-t003:** PLDEs functions of different species and applications of PLDEs in anti-inflammatory substances.

Source	Exosomes	Targeted Cells or Organ	Function	Reference
Ginger	Ginger exosome-like nanoparticle microRNA (miRNA aly-miR396a-5p).	Lung epithelial cells and lung macrophages	Inhibition of TNF-α, IL-6 and IL-1β production as well as NF-κB	[69]
Chinese herbal decoction	PGY-sRNA-6	Poly(I:C)-induced lung inflammation	Inhibition of inflammatory cytokines	[70]
Ginger	Shogaol in the ginger-derived nanoparticle	Mouse hepatocytes	Activation of Nrf2 through the TLR4/TRIF pathway.	[76]
Garlic	Garlic-derived nanovesicles	Liver HepG2 cells and their CD98 receptors	Reduction in the levels of hepatitis-associated TNF-α, IFN-γ and IL-6.	[44]
Ginger Rhizome	Food-borne exosome-like nanoparticles	Liver macrophages	Inhibition of IL-1β and IL-18 secretion and the assembly and activation of NLRP3 inflammasome.	[78]
Shiitake Mushroom	Shiitake mushroom-derived ELNs (S-ELNs)	Inflammasome in primary liver macrophages	Inhibition of NLRP3 inflammasome activation and reduction of Pro-IL-1β and NLRP3 in liver; inhibition IL-6 secretion.	[79]
Ginger	Edible plant derived exosome-like nanoparticles	Intestinal macrophages	Induction of intestinal Nrf2 activation and Wnt/TCF4; increased secretion of IL-10 and HO-1.	[83]
Ginger	Nanoparticles derived from edible ginger	Colonic epithelial cells and macrophages	Increased survival and proliferation of IECs in a colitis model; decreased expression of TNF-α, IL-6, and IL-1β; increased expression of IL-10 and IL-22.	[61]
Grapefruit	Grapefruit-derived nanovesicles	Intestinal macrophages	Upregulation of HO-1 expression; inhibition IL-1β and TNF-α production in intestinal macrophages.	[85]
Grape	Grape exosome-like nanoparticles	Mouse intestinal stem cells	Penetration of the intestinal mucus barrier and promoting the proliferation of intestinal stem cells, regeneration of mucosal epithelia and rapid recovery of the intestinal structure.	[38]
Turmeric	Turmeric-derived nanoparticles	Colonic epithelial cells and macrophages	Inhibition TNF-α, IL-6 and IL-1β expression, upregulation HO-1 expression and inactivation of the NF-κB pathway.	[87]
Tea	Tea leaf-derived natural	Intestinal macrophages	Reduction of ROS, inhibition of pro-inflammatory cytokine expression and upregulation of IL-10 secreted by macrophages.	[82]
Ginger	GELN mdo-miR7267-3p	*Lactobacillus* and *Lactobacillus rhamnosus*	Induction of IL-22 production.	[39]
Mulberry Bark	Exosome-like nanoparticles derived from edible mulberry bark	Virulent Listeria monocytogenes	Inhibition of bacterial mRNA expression and species-specific growth inhibition of the *Listeria* monocytogene strain, EGD.	[88]
Ginger	Ginger exosome-like nanoparticles	HBP35 (Hemopexin 35), *Porphyromonas gingivalis*	Inhibition of *Porphyromonas gingivalis* growth and the associated inflammation.	[36]
Blueberry	Blueberry-derived exosome-like nanoparticles	Endothelial cell line EA.hy926 cells	Inhibition of TNF-α-induced reactive ROS production and alleviates loss of cell viability.	[89]

## Data Availability

Not applicable.

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
