# Peer review of "Advances in the Therapeutic Applications of Plant-Derived Exosomes in the Treatment of Inflammatory Diseases"

_biomedicines, 2023, doi:10.3390/biomedicines11061554_

Round 1

Reviewer 1 Report

Plant-derived exosomes (PLDE) are used e.g. in the treatment of inflammation. The authors focused on this aspect in their review article. Overall, the article is well written. However, authors should indicate in an additional subsection:
a) the greatest advantages of treatment with PLDE;
b) the greatest disadvantages (side effects) of using it for the treatment of PLDE;
c) further prospects for PLDE applications.

Author Response

Response to Reviewer 1 Comments

Plant-derived exosomes (PLDE) are used e.g. in the treatment of inflammation. The authors focused on this aspect in their review article. Overall, the article is well written. However, authors should indicate in an additional subsection:

  1. a) the greatest advantages of treatment with PLDE;
  2. b) the greatest disadvantages (side effects) of using it for the treatment of PLDE;
  3. c) further prospects for PLDE applications.

Response a: The authors added to the fourth subsection of the manuscript about the disadvantages of PLDEs. Here is the original content:” The biggest advantage of PLDE as a therapeutic agent is that it is rich in types, diverse in functions, has excellent engineering transformation potential as carrier.

Response b: The authors added to the fourth subsection of the manuscript about the disadvantages of PLDEs.

Response c: The authors added to the 5th subsection of the manuscript about the disadvantages of PLDEs.

Reviewer 2 Report

General comments:

This review summarizes the biogenesis, components, and functions of PLDEs, and focuses on their use as therapeutic agents in the treatment of inflammatory diseases.

Minor comments:

1. Figures were suggested to enlarge and make it clearer to read.

Author Response

Response to Reviewer 2 Comments

General comments:

This review summarizes the biogenesis, components, and functions of PLDEs, and focuses on their use as therapeutic agents in the treatment of inflammatory diseases.

Minor comments:

  1. Figures were suggested to enlarge and make it clearer to read.

Response Minor comments : In response to your comments, the authors have made adaptive revisions based on the editorial rules of the journal

Reviewer 3 Report

According to the data provided, the potential benefits of using exosomes from plants in many inflammatory diseases and cancer in humans are amazing.

At the same time, the question arises whether they do not also have a potential danger, about which the study does not provide data.

The potential danger could come, in my opinion, exactly through the manipulation of the plant exosomes in the sense of creating specialized vehicles  to carry genetic (RNA) and immunogenic (proteins) information into the human body.

This way, the question can be asked if the exosomes could not be the cause of food allergies and intolerances, precisely because they introduce proteins into the bloodstream. Also, the human digestive system is set to decompose/digest such type of entities, with the help of acid, enzymes and bile secretion. Therefore, naturally, human body seems to oppose to their entry into circulation. The more so as data presented indicate that they are not present in normal case of preparing/using plant derived products and once allowed, they can induce major effects in human organs <The study about exosomes of shiitake mushrooms shows that ELNs from various common mushroom species, but found that only shiitake-derived ELNs (S-ELNs) significantly inhibited NLRP3 by preventing inflammasome formation in primary macrophages. Inflammasome activation, which occurs via a different mechanism to lentinan, it inhibits inflammation by targeting and inhibiting AIM2 inflammasome-associated activity.>

Altogether, the information provided by the study are very interesting and of huge importance for many diseases, especially on IBD, but should also be added a section of potential dangerous effects by using them.

Author Response

Response to Reviewer 3 Comments

According to the data provided, the potential benefits of using exosomes from plants in many inflammatory diseases and cancer in humans are amazing.

At the same time, the question arises whether they do not also have a potential danger, about which the study does not provide data.(point 1)

The potential danger could come, in my opinion, exactly through the manipulation of the plant exosomes in the sense of creating specialized vehicles  to carry genetic (RNA) and immunogenic (proteins) information into the human body.

This way, the question can be asked if the exosomes could not be the cause of food allergies and intolerances, precisely because they introduce proteins into the bloodstream.(point 2) Also, the human digestive system is set to decompose/digest such type of entities, with the help of acid, enzymes and bile secretion. Therefore, naturally, human body seems to oppose to their entry into circulation. The more so as data presented indicate that they are not present in normal case of preparing/using plant derived products and once allowed, they can induce major effects in human organs <The study about exosomes of shiitake mushrooms shows that ELNs from various common mushroom species, but found that only shiitake-derived ELNs (S-ELNs) significantly inhibited NLRP3 by preventing inflammasome formation in primary macrophages. Inflammasome activation, which occurs via a different mechanism to lentinan, it inhibits inflammation by targeting and inhibiting AIM2 inflammasome-associated activity.>

Altogether, the information provided by the study are very interesting and of huge importance for many diseases, especially on IBD, but should also be added a section of potential dangerous effects by using them.(point 3)

Response 1: There are very few negative reports on studies on PLDEs, and it can hardly find precise data presented in this manuscript.

Response 2: There are no relevant studies confirming that PLDEs can be the cause of food allergies and intolerances.

Response 3: The authors added to the fourth subsection of the manuscript about the disadvantages of PLDEs.